# Understanding and Comprehensive Evaluation of Cold Resistance in the Seedlings of Multiple Maize Genotypes

**DOI:** 10.3390/plants11141881

**Published:** 2022-07-20

**Authors:** Xiaoqiang Zhao, Cai Zhao, Yining Niu, Wun Chao, Wei He, Yifan Wang, Taotao Mao, Xiaodong Bai

**Affiliations:** 1State Key Laboratory of Aridland Crop Science, Gansu Agricultural University, Lanzhou 730070, China; zhaoxq3324@163.com (X.Z.); zhaoc@gsau.edu.cn (C.Z.); hew@gsau.edu.cn (W.H.); wangyf@gsau.edu.cn (Y.W.); m15294392789@163.com (T.M.); bxd15293898130@163.com (X.B.); 2United States Department of Agriculture, Fargo, ND 58102-2765, USA; wun.chao@usda.gov

**Keywords:** maize, cold resistance, reactive oxygen species level, membrane stability, antioxidant enzyme activity, chlorophyll content, photosynthetic performances, polyamines metabolism, gene expressions

## Abstract

Maize is a cold-sensitive crop, and it exhibits severe retardation of growth and development when exposed to cold snaps during and right after seedling emergence. Although different agronomic, physiological, and molecular approaches have been tried to overcome the problems related to cold stress in recent years, the mechanisms causing cold resistance in maize are still unclear. Screening and breeding of varieties for cold resistance may be a sustainable option to boost maize production under low-temperature environments. Herein, seedlings of 39 different maize genotypes were treated under both 10 °C low temperature and 22 °C normal temperature conditions for 7 days, to assess the changes in seven growth parameters, two membrane characteristics, two reactive oxygen species (ROS) levels, and four antioxidant enzymes activities. The changes in ten photosynthetic performances, one osmotic substance accumulation, and three polyamines (PAs) metabolisms were also measured. Results indicated that significant differences among genotypes, temperature treatments, and their interactions were found in 29 studied traits, and cold–stressed seedlings were capable to enhance their cold resistance by maintaining high levels of membrane stability index (66.07%); antioxidant enzymes activities including the activity of superoxide dismutase (2.44 Unit g^−1^ protein), peroxidase (1.65 Unit g^−1^ protein), catalase (0.65 μM min^−1^ g^−1^ protein), and ascorbate peroxidase (5.45 μM min^−1^ g^−1^ protein); chlorophyll (Chl) content, i.e., Chl a (0.36 mg g^−1^ FW) and Chl b (0.40 mg g^−1^ FW); photosynthetic capacity such as net photosynthetic rate (5.52 μM m^−2^ s^−1^) and ribulose 1,5–biphosphate carboxylase activity (6.57 M m^−2^ s^−1^); PAs concentration, mainly putrescine (274.89 nM g^−1^ FW), spermidine (52.69 nM g^−1^ FW), and spermine (45.81 nM g^−1^ FW), particularly under extended cold stress. Importantly, 16 traits can be good indicators for screening of cold–resistant genotypes of maize. Gene expression analysis showed that *GRMZM2G059991*, *GRMZM2G089982*, *GRMZM2G088212*, *GRMZM2G396553*, *GRMZM2G120578*, and *GRMZM2G396856* involved in antioxidant enzymes activity and PAs metabolism, and these genes may be used for genetic modification to improve maize cold resistance. Moreover, seven strong cold–resistant genotypes were identified, and they can be used as parents in maize breeding programs to develop new varieties.

## 1. Introduction

Maize (*Zea mays*), originating from tropical and subtropical regions, is extremely sensitive to cold stress, particularly at the seedling emergence stage [1]. Cold damage during maize seedlings development in early spring usually occurs at suboptimal temperatures of 10–15 °C or at even lower temperatures of 2–8 °C [2], which normally results in low seedling emergence/uniformity and biomass production, growth retardation, as well as occasionally irreversible tissue damage [3,4,5]. The growth of cold-sensitive maize is limited in Northeast China, high–latitude areas since a transient chilling exposure, on average, every 3–4 years, may cause yield losses of 20% to 30% [1,6]. In agricultural practice, demand is high for the development of varieties with improved cold–resistance for the longer growing season, so that yield potentials can be fully realized in high–latitude cultivation regions.

Significant progress has been made in understanding cold–resistance mechanisms in maize. Low-temperature pressure caused lipid peroxidation, DNA damage, protein denaturation, carbohydrate oxidation, pigment decomposition, and even subsequent cell death in maize seedlings through excessive production of reactive oxygen species (ROS; including O_2_^•−^, OH^−^, and H_2_O_2_) [3,4]. ROS production also damaged the chloroplast, mitochondria, and metabolic enzymes [4,5]. The pale green appearance is caused by low levels of chlorophyll (Chl) a and Chl b, resulting in decreased electron transport in the photosystems, the activity of the enzymes involved in carbon fixation including 1,5–biphosphate carboxylase (Rubisco), NADP-malic enzyme (NADP–ME), pyruvate phosphate dikinase (PPDK), and phosphoenolpyruvate carboxykinase (PEPCK), and net photosynthesis efficiency [5]. Cold temperatures also affected the abundance of carotenoids, which enhanced photoprotection by promoting the dissipation of excitation energy [3]. In addition, maize seedlings evolved an efficient antioxidant system including superoxide dismutase (SOD), peroxidase (POD), catalase (CAT), glutathione peroxidase (GPX), and ascorbate peroxidase (APX) to prevent extremely ROS accumulation under low-temperature stress [4,7]. Consequently, enhanced levels of different antioxidant gene transcripts have been found in maize as a potential low-temperature stress defense strategy. As an important amino acid and osmotic adjustment substance, proline (Pro) was involved in response to various environmental stresses to prevent cell dehydration [8]. The rapid accumulation of Pro was the initial reaction of plants exposed to the cold condition [9]. Interestingly, previous studies also reported that polyamines (PAs) were involved in several physiological and developmental processes in plants, such as survival of plant embryos and translation [10], cell proliferation [11], modulated gene expression [12], membrane stabilization [13], and chilling stress response [11]. In recent years, only a few studies have shown that increasing PA concentrations in maize [12,13,14] could improve its resistance to low-temperature stress. However, there are still many questions that remain unanswered. These questions, for instance, include: which physiological changes and metabolic disturbances are more important to maize after low-temperature injury; what are the internal relations among different physiological metabolisms; and how they act synergistically to form maize cold resistance potential.

Although maize originated in a tropical and subtropical regions, genetic variation in cold–resistance exists in maize germplasm. For example, Leipner [15] reported that European flint and highland tropical germplasm materials were more chilling–resistant than dent material from the Corn Belt. The major challenge that maize breeders face is the complexity of low-temperature responses because the acclimation of maize to low temperature is a complex process. It should be noted, however, that the intraspecific diversity of maize underlies its excellent adaptability to contrasting climates, and the cold-resistant cultivars were obtained after subjecting maize to prior low-temperature acclimation [16]. Thus the testing of maize genotypes for divergent cold–resistance in terms of their physiological and biochemical responses to low-temperature stress may serve as a potent approach to screen and developing new cultivars [2,5,17]. In addition, evaluating the physiological and biochemical changes occurring under low-temperature stress may lead to the genetic improvement of cold–resistant genotypes and be beneficial for their elite allele exploration and application in germplasm resources [16,17]. We hypothesize that (i) low temperature caused maize physiological changes and metabolic disturbances, and (ii) they act synergistically to form maize cold resistance. In the context of these facts, this study aimed to explore the response mechanism of multiple agronomic and physiological traits of 39 elite maize genotypes seedlings under 22 °C and 10 °C temperature conditions, establish a reliable maize screening method for large–scale selection or for breeding stock with cold–resistance, and further screen maize genotypes for resistance against low-temperature stress in spring maize region of Northwest China. In addition, the expression level of associated genes was determined via reverse–transcription quantitative PCR (RT–qPCR). These findings will lay a foundation for developing excellent maize cultivars with cold resistance.

## 2. Results

### 2.1. Combined Analysis of Temperature Treatments and Maize Genotypes on All Traits

Factorial analysis of variance (ANOVA) proved significant for the tested variables (maize genotype and temperature) and their physiological and molecular interactions on the 29 tested traits including seven growth parameters, two membrane characteristics, two ROS levels, four antioxidant enzymes activities, ten photosynthetic performances, one osmotic adjustment substance accumulation, and three PAs metabolisms in 39 maize genotypes seedlings (Table 1). This data demonstrated that the resistance to low-temperature stress was controlled by the factor of maize’s own genetic constitution, ambient temperature, and interaction. Our results also showed that the temperature imposed higher influences on these traits.

### 2.2. Maize Genetic Variation of All Traits

The genetic differences were confirmed by the genetic variation coefficient (CV_g_) of 29 tested traits among 39 maize genotypes when the temperature was controlled at the seedling stage (Table 1). The CV_g_ ranged from 4.59% [spermine (Spm) concentration] to 37.59% (CAT activity) under the normal temperature of 22 °C and from 8.43% [water use efficiency (WUE)] to 40.13% (CAT activity) under low-temperature stress of 10 °C for the 29 tested traits. We hypothesized that there may be higher broad sense heritability for 22 tested traits including seedling length, root length, root number, seedling fresh weight, root fresh weight, plant biomass, root–shoot ratio, Chl a content, Chl b content, Chl a+b content, net photosynthetic rate (Pn), ribulose Rubisco activity, malondialdehyde (MDA) content, O_2_^•−^ production rate, H_2_O_2_ content, SOD activity, POD activity, CAT activity, APX activity, Pro content, spermidine (Spd) concentration, and Spm concentration since their CV_g_ were above 15% in one or two temperature environments.

### 2.3. Maize Phenotypic and Physiological Variations of All Traits in Response to Low-Temperature Stress

Seedlings of 39 maize genotypes were subjected to low-temperature stress (10 °C) for 7 days, and the results revealed that seedlings’ growth and development were in general significantly inhibited (Figure 1). Compared with normal temperature treatment of 22 °C, 10 °C caused a significant decrease of 48.6% in seedling length, 39.0% in root length, 10.2% in root number, 28.1% in seedling fresh weight, 21.1% in root fresh weight, and 24.2% in plant biomass, respectively; however, a prominent increase of 9.9% in root–shoot ratio was observed (Figure 2). These data indicated that the 10 °C low-temperature stress mainly occurred on aboveground tissues of maize seedlings, in particular on young leaves.

Chlorophyll is an essential cofactor for photosynthesis, and maintaining a normal chlorophyll state and level is essential for photosynthetic efficiency and carbon fixation, which directly influence plant growth and development under various abiotic stresses [18]. Comparing seedlings of 39 maize genotypes exposed to 10 °C low-temperature stress with 22 °C control, 10 °C significantly reduced the contents of Chl a, Chl b, Chl a+b in their leaves by 18.6%, 21.7%, and 20.3%, respectively; however, chlorophyll a:b ratio (Chl a/b) slightly increased by 3.3% (Figure 1 and Figure 2). The data suggested that photosynthesis is highly sensitive to cold stress [19]; cold stress specifically inhibited chlorophyll biosynthesis in leaves of maize seedlings, and the levels of Chl b were lower than Chl a. Under low-temperature stress, the Pn, stomatal conductance (Gs), transpiration rate (Tr), WUE, and Rubisco activity in leaves of 39 maize genotypes seedlings were 19.7%, 12.0%, 2.8%, 17.1%, and 25.1% lower than that of untreated seedlings; in contrast, intercellular CO_2_ (Ci) was 7.2% higher in cold–stressed seedlings relative to untreated seedlings (Figure 1 and Figure 2). These data suggested that cold–induced reduction of photosynthesis in maize seedlings was primarily caused by CO_2_ diffusion from sub–stomatal interval cavities to the carboxylation site in chloroplasts and the degree of stomatal closure. The results also explained why Rubisco activity and Gs decreased significantly in different maize genotypes accessions under low-temperature stress.

Plant MDA content and membrane stability index (MSI) reflect the degree of lipid peroxidation and membrane stability under various environmental stresses [20,21]. Cold imposition in leaves of 39 maize genotypes seedlings significantly increased MDA content a 221.9% was observed relative to control temperature (Figure 1 and Figure 2). In contrast, the low temperature decreased MSI value by 19.5% in leaves of 39 maize genotypes seedlings (Figure 1 and Figure 2). In addition, Pro is a key osmoregulation substance that could resist dehydration and reduce cellular damage caused by cold stress [19]. In this study, the leaves’ Pro content significantly increased by 311.8% in 39 maize genotypes seedlings after low-temperature stress treatments (Figure 1 and Figure 2). This data implied that cold stress could promote the accumulation of Pro in maize leaves, thus improving maize cold stress resistance.

Excess energy and electrons can mediate ROS production under cold stress; however, plants often alleviate peroxidation damage caused by ROS by increasing antioxidant enzymes [19]. After low-temperature treatment, leaves O_2_^•−^ production rate increased by 163.4% in seedlings of 39 maize genotypes compared with the control (Figure 1 and Figure 2). Similarly, leaves H_2_O_2_ production increased by 159.2% in these maize seedlings (Figure 1 and Figure 2). These results suggested that excessive ROS production may be the primary cause of cold damage in maize leaves. In addition, low-temperature stress significantly increased the activities of SOD (42.7%), POD (144.4%), CAT (101.1%), and APX (65.8%) in leaves of 39 maize genotypes seedlings (Figure 1 and Figure 2). These data showed that cold stress could enhance antioxidant enzyme activities under severe stress, thus improving maize cold tolerance.

PAs are small aliphatic low–molecular weight polycationic nitrogenous compounds that played critical roles in chilling tolerance in plants [12,13,14]. Upon exposure to low temperature, the concentrations of putrescine (Put), Spd, and Spm significantly increased by 183.7%, 69.2%, and 56.6% in leaves of 39 maize genotypes seedlings, respectively (Figure 1 and Figure 2). The data indicated that maize seedlings could accumulate different PAs under low-temperature stress, which may be closely associated with cold resistance in maize.

### 2.4. Framework for Relationships among All Traits

To better understand the framework of internal relations among 29 tested traits and cold resistance potential of 39 maize genotypes, Pearson pairwise correlation analysis and principal component analysis (PCA) were conducted under both temperature treatments (Figure 3a–d). Pearson pairwise correlation analysis revealed that there were 181 groups with significant (*p* < 0.05 or *p* < 0.01) correlation between both traits (Figure 3a,b). This data suggested that the inherent relationships among multiple traits were complex under different temperature conditions, which caused the information to overlap; thus, it was appropriate to conduct PCA analysis. Based on the eigenvalues and cumulative variance for all traits, the first five principal components (PCs; PC1, PC2, PC3, PC4, and PC5) accounted for 91.152% of the total variance, of which the eigenvalues were larger than 1.0. These PCs were linear combinations of the different traits based on their variable loadings (Figure 3c,d). Specifically, seedling length, root length, seedling fresh weight, root fresh weight, plant biomass, Chl a content, Chl a content, Chl a+b content, Pn, Ci, Gs, WUE, Rubisco activity, MDA content, and MSI were the primary traits in PC1, which accounted for 49.683% of the total variance. Tr, O_2_^•−^ production rate, H_2_O_2_ level, SOD activity, POD activity, CAT activity, APX activity, Pro content, Put concentration, Spd concentration, and Spm concentration were the primary traits in PC2, which accounted for 24.528% of the total variance. Root–shoot ratio was the primary trait in PC3, which accounted for 8.749% of the total variance. Chl a/b ratio was the primary trait in PC4, which accounted for 4.325% of the total variance. Root number was the primary trait in PC5, which accounted for 3.867% of the total variance. Therefore, these traits jointly determine the cold resistance of maize seedlings through their synergistic or inhibitory effects.

### 2.5. Comprehensive Evaluation of Cold Resistance among All Maize Genotypes Seedlings

The cold resistance coefficient (CRC), is a measure of cold stress effects based on the reduction of each trait under cold stress treatments in comparison to un–stress treatments, and thereby was used for the identification of cold–resistant genotypes [22]. The membership function value of cold resistance (MFVC), calculated from CRC, provided a comprehensive evaluation method for cold resistance of genotypes based on multi-traits determination [23]. In these regards, the data obtained from the two temperature experiments were used to evaluate maize cold resistance at the seedling stage. We first calculated the cold resistance coefficient from each trait among 39 maize genotypes seedlings (Figure 4a). The values were then used to access the average value of the membership function of measured traits for each maize genotype for cold resistance (U_ij_). The results showed that the cold resistance (U_j_) of 39 maize genotypes ranged from 0.152 (M1012 genotype) to 0.743 (M0504 genotype), and their coefficient of variation was 34.2% (Figure 4b). Therefore, there were significant differences in cold resistance among maize genotypes. In addition, the diversities of U_ij_ across 39 maize genotypes were visualized by hierarchical clustering and heatmap. As shown in Figure 4b, these maize genotypes were divided into four types, i.e., type I included seven strong cold–resistant maize genotypes (U_j_, 0.656–0.743), type II included six moderate cold–resistant genotypes (U_j_, 0.574–0.638), type III included seven weak cold–resistant genotypes (U_j_, 0.411–0.478), and type IV included 19 cold–sensitive genotypes (U_j_, 0.152–0.393), respectively (Figure 4b).

### 2.6. Differential Expression of Six Candidate Genes Involved in ROS Scavenging and PAs Metabolisms

The expression of six candidate cold–induced genes including *arginine decarboxylase 1 (ADC1*; *GRMZM2G396553*), *spermidine synthase 1* (*SPDS1*; *GRMZM2G120578*), *probable polyamine oxidase* 2 (*PAO2*; *GRMZM2G396856*), *superoxide dismutase 3* (*SOD3*; *GRMZM2G059991*), *peroxidase 72* (*POD72*; *GRMZM2G089982*), and *catalase 1* (*CAT1*; *GRMZM2G088212*) were examined in maize seedlings of M1012 (cold–sensitive genotype) and M0504 (strong cold–resistant genotype) after treated these seedlings with 10 °C and 22 °C for 7 days. These genes were associated with PAs metabolisms and antioxidant enzymes system and were known to play a role under cold stress. The results showed that the relative expression levels of these six genes were significantly increased in leaves of both maize genotypes seedlings after low temperature treatment (10 °C) (Figure 5).

## 3. Discussion

Low temperature is among the major abiotic stresses that limited the productivity, quality, as well as distribution of maize [23]. Generally, tropical and subtropical maize are sensitive to cold stress [1]; therefore, maize that resists cold spells in early spring could broaden planting areas in high latitude regions, allow crop rotation, and increase yield by extending the growing season [24]. Moreover, the studies of different crops have shown that cold resistance is a complex biological process that involves interactions between agronomic traits and physiological processes and is dependent on the level of cold severity and timing in relation to the stage of crop development [7,25,26,27]. Therefore, it is vital for maize breeding programs to understand the physiological changes during seed germination and seedling development under low-temperature stress.

It is well understood that low temperature affected maize seed germination, early leaf development, and seedling growth [28]; leaf sheath and mesocotyl became brown, and leaves wilted and showed symptoms of waterlogging when the damage became more serious [29]. Maize leaves develop under relatively mild cold temperatures (10–15 °C) might have a pale green chlorotic appearance, and leaves become yellow when exposed to 6 °C or lower cold temperatures [30]. Cold stress of 5 °C could even lead to premature leaf senescence or seedling death [31]. Similarly, in this study, the seedlings of 39 maize genotypes were exposed to 10 °C low temperature resulted in retarded plant development, and seedlings were more sensitive to low-temperature stress than that of roots, i.e., seedling length decreased by 48.6%, root length decreased by 39.0%, root number decreased by 10.2%, seedling fresh weight decreased by 28.1%, root fresh weight decreased by 21.1%, and plant biomass decreased by 24.2% under 10 °C low-temperature stress, respectively; which may be caused by the changes of various physiological processes.

It is also worth noting that the generation of ROS and the balance between the production and scavenging of ROS under low-temperature stress led to different cellular signaling events dictate cellular senescence, apoptosis, and cellular signaling, or as a stress-induced signal involved in the biosynthesis of some secondary metabolites such as anthocyanin, phytohormones, and lignin [32,33,34]. Our findings were also consistent with an earlier study [32], namely, 10 °C environments significantly increased H_2_O_2_ (159.2%) and O_2_^•−^ (163.4%) levels in leaves of 39 maize genotypes seedlings. Excessive ROS production may be the main factor of membrane lipid peroxidation in the leaves of maize seedlings during chilling damage. Moreover, ROS promoted the degradation of polyunsaturated fatty acids and the production of MDA [35]. MDA has long been used as a lipid peroxidation marker in studies related to oxidative stress and redox signaling, particularly in studies focused on plant responses to various environmental stresses, in which MDA accumulation caused damage to cell membrane system [36]. In this study, 10 °C low temperature affected MSI and MDA content in leaves of all 39 maize genotypes seedlings; MSI value (19.5%) was clearly decreased; however, MDA content (221.9%) was markedly increased after 10 °C low-temperature treatment, suggesting that cellular membrane damage was induced by low-temperature stress.

However, plants have antioxidant enzymes to counteract the effect of ROS; the increase of SOD, POD, CAT, and APX activities could scavenge excessive ROS [4,7]. Increasing evidence has shown that low temperature induced activation of antioxidant enzyme activities in *Nicotiana tabacum* L. and *Secale cereale* L. [37]. This phenomenon also has been observed in leaves of our 39 maize genotypes seedlings under 10 °C low-temperature stress, the activities of SOD, POD, CAT, and APX were significantly increased by 42.7–144.4%. In addition, Ramazan et al. [38] reported that the expressions of *APX*, *SOD*, *CAT*, and *POD* genes were correlated well with their activities for maize in different environments. Dreyer and Dietz [39] showed that *APX2* gene in *Arabidopsis thaliana* L. was up-regulated 2.4–fold after 30 min and maximally 5.6–fold after 6 h of cold treatment. We also analyzed the expression of three candidate genes encoding antioxidant enzymes, namely *GRMZM2G059991* (*SOD3*), *GRMZM2G089982* (*POD72*), and *GRMZM2G088212* (*CAT1*), and the results showed that the expression of these genes was significantly increased in both cold–sensitive M1012 and strong cold–resistant M0504 genotypes under 10 °C stress compared with 22 °C control; however, their transcript abundance was considerably greater in M0504 than that of M1012 at 10 °C.

For overwinter plants or plants living in cold regions, the exposure of plants to freezing stress can lead to the formation of ice crystals in the extracellular space of plant cells, and large ice crystals can physically damage the plasma membrane [40]. Interestingly, as one of the osmotic cryoprotectants, during the freezing process, Pro could penetrate through the cell membrane quickly, bind intracellular water hydrogen bonds, inhibit the formation of ice crystals at freezing temperatures, and protect protein integrity, and cell membranes [41]. Our results showed a significant increase in Pro levels (311.8%) in maize leaves, which may be another crucial adaptive strategy to minimize the negative impacts of low-temperature stress. In the future, the spatiotemporal transcription level of key genes responsible for Pro biosynthesis enzyme, especially *pyrroline**–5–carboxylate synthase* (*P5CS*) in maize under cold stress remains to be further studied. Furthermore, the exposure to low temperature promoted PAs accumulation in maize, *Oryza sativa* L., and *Cicer arietinum* L. [13]. PAs could function as stress messengers in plant responses to different stress signals. Gao et al. [13] pointed out that PAs were a source of ROS synthesis due to that their catabolism produced a strong oxidizer H_2_O_2_ that activates an antioxidant defense response to various abiotic stresses. PAs play several important roles in plant survival; PAs can bind to the phospholipid site of the cell membrane to maintain membrane stability, increase the activities of SOD, POD, and CAT to scavenge ROS, protect normal cell permeability, and prevent water loss. Our study correlated well with these studies as 10 °C stress significantly promoted Put, Spd, and Spm accumulation (56.6–183.7%) in leaves of maize seedlings. Gao et al. [12] reported that *ZmADC1*, *ZmADC2*, *ZmODC*, *ZmSPDS*, *ZmSAMDC2*, *ZmPAO1*, *ZmPAO2*, and *ZmPAO3* responded to biosynthesis and catabolism of PAs in maize under cold stress, which was also consistent with the expressions of *GRMZM2G396553*, *GRMZM2G120578*, and *GRMZM2G396856* in M0504 and M1012 under 10 °C and 22 °C treatments. Thereby, these results provide a molecular basis for the utilization of PA–related genes to improve maize cold resistance in the future.

Photosynthesis is sensitive to changes in environmental stimuli. Under low-temperature stress, the ultrastructure of chloroplast in *Brassica campestris* L. was damaged, which caused the block of chlorophyll biosynthesis while accelerating chlorophyll degradation [42]. This phenomenon is consistent with our current observation in maize leaves; the levels of Chl a, Chl b, and Chl a+b were significantly decreased under 10 °C stress, resulting in the significant inhibition of photon acquisition, light conversion, electron transfer, and these inhibitions varied greatly between cold-resistant and cold-sensitive maize genotypes. Gilmore [43] found that low temperature increased the abundance of carotenoids and xanthophyll carotenoids relative to Chlorophyll. Zeaxanthin (a common carotenoid alcohol) is involved in dissipating excess excitation energy that cannot be used for photosynthesis, Pfündel and Bilger [44] reported that cold stress also resulted in the de–epoxidation of xanthophyll cycle pigments by converting violaxanthin to zeaxanthin via antheraxanthin. Therefore, cold–hardened cereals and cold–tolerant plants accumulated antheraxanthin and zeaxanthin a cold environment [30]. In addition, if the ambient temperatures dropped significantly below growth temperature, an imbalance between photosynthetic light and dark reactions was established, and the photosynthetic electron transport chain released more O_2_^•−^ by transferring electrons to molecular oxygen [39], resulting in poor photosynthetic performance. This poor photosynthetic performance was associated with photo-damage to the photosystem II (PSII) reaction centers, alterations in pigment composition of leaves, increased dissipation of excess energy in the antennae of PSII, lower activities of enzymes in the carbon cycle, and altered chloroplast development [17]. Liu et al. [42] reported that the decrease in leaves Pn of *Brassica campestris* L. was due to the functional disorder of the photosynthetic apparatus and blocked chlorophyll synthesis at diurnal/nocturnal temperatures of 20/−5 °C; however, the increase in leaves Ci was mainly due to stomatal closure. Rubisco is involved in the photosynthetic carbon reduction cycle. Rubisco activase (RCA) is an important enzyme that catalyzes the carboxylation and oxygenation of Rubisco. Bi et al. [45] found that overexpression of *Cucumis sativus* L. *CsRCA* gene improved its growth and low-temperature tolerance. Our results indicated that low-temperature stress decreased leaves Pn, Gs, Tr, WUE, Rubisco by 2.8–25.1%, while increased Ci by 7.2% in maize seedlings, which was consistent with maize photosynthetic performances in response to drought tolerance [15]. Thus, these studies illustrated that the changes in photosynthetic performances may be related to low-temperature stress intensity and genotypic adaptability.

According to the results of previous studies [1,2,3,4,5,6,12,13,14,16,17,23,28,38,39,46] and our Pearson correlation and PCA analyses in the current study, distinct physiological processes and organ systems were involved in the onset, accumulation, and/or repair of maize cold injury; concomitantly, distinct molecular mechanisms were also likely involved in preventing (i.e., resistance) versus repairing cold injury (i.e., tolerance) by activating and inducing the expressions of some cold-responsive genes [47]. A diagram of hypothetical complex response mechanisms on cold resistance in maize was constructed (Figure 6). Briefly, maize seedlings were grown in low-temperature environments. Cold temperature activated the expression of multiple genes responsible for antioxidant enzymes and PAs pathway, which then altered multiple physiological processes such as ROS homeostasis, lipid peroxidation level, osmotic adjustment ability, antioxidant enzymes system, chlorophyll biosynthesis, photosynthetic capacity, and PAs deposition level in leaves. Concomitantly, these physiological reactions also interacted with each other and further activated secondary and tertiary regulatory networks to prevent or reduce the adverse effects of cold stress.

In addition, the comprehensive evaluation of cold resistance among all 39 maize genotypes seedlings using membership function value and hierarchical clustering analysis indicated that the strong cold–resistant maize genotypes were enhanced in many traits including MSI value, CAT activity, POD activity, Chla content, Chl b content, Chl a+b content, Pn, Rubisco activity, Tr, WUE, Put concentration, Spd concentration, Spm concentration, seedling length, seedling fresh weight, and plant biomass; in contrast, these traits were extremely poor for the cold-sensitive maize genotypes. Furthermore, the CV_g_ of these traits was larger under different temperature treatments. Zhao et al. [48] reported that the broad sense heritability ($h^{2}$) of Pn, Tr, Rubisco activity, and WUE were 88.9%, 69.7%, 68.8%, 86.8% in a maize F_4_ population of 218 lines under well-watered and water-stress environments, respectively, while their genotype × environment interaction heritability ($h_{ge}^{2}$) was 9.9%, 15.6%, 9.7%, and 1.5%, respectively. Goering et al. [49] showed that the $h^{2}$ of chlorophyll concentration, i.e., absorbance at 663 nm (Chl a) and 645 nm (Chl b) were 88.5% and 82.5% in a maize IBM recombinant Inbred Lines (RIL) population at 24 °C and 4 °C conditions, respectively. David et al. [50] analyzed the genetic basis of SOD activity, CAT activity, POD activity, APX activity, root length, and root weight by QTL mapping in rice. Wassom et al. [51] also identified the QTLs and allelic genes responsible for polyamine oxidase in maize in another culture. These results indicated that the above 16 traits were greatly regulated by maize’s own genetic basis under various environmental stress; thus, these traits could serve as reliable indicators or valuable standards for evaluating the cold resistance of maize. Similarly, a banana variant, the *Musa nana* Lour. Williams, showed more cold tolerance than Grand Nain, a *Musa acuminate* cultivar, by accumulating higher levels of the photosynthetic pigments, which could be utilized as important selectable indicators for screening cold tolerant banana genotypes [52]. Meanwhile, we identified that M0504 (U_j_, 0.743), F0305 (U_j_, 0.700), M02021 (U_j_, 0.680), M1017 (U_j_, 0.678), F0801 (U_j_, 0.659), F1230 (U_j_, 0.658), and lg104-1 (U_j_, 0.656) were the strongest resistant germplasm to low temperature stress in this study, which will be useful materials for the development of new cold-resistant varieties in maize.

## 4. Materials and Methods

### 4.1. Maize Genotypes and Different Temperature Treatments

The 39 elite maize genotypes were the core germplasm resources of spring maize in Northwest China, provided by Crop Research Institute, Gansu Academy of Agricultural Sciences, China (https://zws.gsagr.cn/, accessed on 16 April 2022). Seeds were surface sterilized in 0.5% (*v*/*v*) sodium hypochlorite for 10 min, followed by three rinses with double–distilled water. Seeds were soaked in double–distilled water for 24 h, sown in sterile sand with moisture content of 20% in a plastic box, and incubated for 15 days at 22 °C. These seedlings were further grown for 7 days at 10 °C low temperature and 22 °C normal temperature treatments, respectively. For all of the growth period, seedlings were cultured in a greenhouse of 12/12 h light/dark cycle, 300 μM m^–2^ s^–1^ light intensity, and 60% relative humidity, and 30 mL Hoagland solution was added to sand at 2-day intervals. Each temperature treatment was done in three replicates with 30 seeds per replicate in this study.

### 4.2. Phenotypic Observations

Six seedlings were selected from 39 maize genotypes seedlings under each temperature treatment, then their seedling length (SL), root length (RL), root number (RN), seedling fresh weight (SFW), root fresh weight (RFW), and plant biomass (PB) were measured, and the root–shoot ratio (RSR) was calculated as follows:RSR = RFW/SFW(1)

### 4.3. MSI and MDA Content

A 0.1 g of leaves was placed in test tubes containing 10 mL of double–distilled water and incubated at 40 °C for 30 min. The conductivity (C_1_) was measured using a DDSJ-308F conductivity meter (Rex Electric Chemical, China). The same set was then kept in a water bath at 100 °C for 15 min, and the conductivity was recorded (C_2_). The MSI [20] was calculated as follows:MSI = [1 − (C_1_/C_2_)] × 100%.(2)

The lipid peroxidation was measured by MDA content [53]. Briefly, a 0.5 g of leaves was homogenized in 5 mL of 0.3% thiobarbituric acid (TBA) and 10% (*v*/*v*) trichloroacetic acid. After incubation at 100 °C for 30 min, mixtures were centrifuged at 12,000 rpm (Centrifuge 5425/5425 R; Eppendorf, Germany) for 10 min. The absorbance of the colored supernatant was measured at 450 nm, 532 nm, and 600 nm, respectively. Then MDA concentration was calculated and its content was expressed as µM g^−1^ fresh weight (FW).

### 4.4. ROS Level

A 0.5 g of leaves were homogenized in 6 mL of phosphate buffer (65 mM, pH 7.8) and then centrifuged at 10,000 rpm (Centrifuge 5425/5425 R; Eppendorf, Germany) at 4 °C for 10 min. Next, 1 mL of supernatant, 0.9 mL of phosphate buffer (65 mM, pH 7.8), and 0.1 mL of hydroxylamine hydrochloride (10 mM) were mixed, and the solutions were placed in a water bath for 20 min at 25 °C. To 0.5 mL of those solutions, 0.5 mL of sulfanilic acid (17 mM) and 0.5 mL of α-naphthylamine (17 mM) were added, followed by 20 min of incubation at 25 °C in a water bath. O_2_^•−^ production rate was determined following the addition of N-butyl alcohol and read at 530 nm, which was expressed as μM min^−1^ g^−1^ FW [54]. A 0.5 g of leaves was homogenized in 5 mL of ice-cold trichloroacetic acid (0.1%, *w*/*v*) and centrifuged at 12,000 rpm (Centrifuge 5425/5425 R; Eppendorf, Germany) at 4 °C for 15 min to collect the supernatant, then optical absorption of the supernatant was measured spectrophotometrically at 410 nm to measure H_2_O_2_ content and expressed as μM g^−1^ FW [55].

### 4.5. Pro Content

A 0.5 g of leaves was homogenized in 10 mL of 3% ice-cold sulfosalicylic acid and then centrifuged at 12,000 rpm (Centrifuge 5425/5425 R; Eppendorf, Germany) at 4 °C for 15 min. The supernatant, 2 mL, was mixed with 2 mL of acid ninhydrin and 2 mL of glacial acetic acid. After incubation at 100 °C for 1 h, 4 mL of toluene was added to extract the red substance. After standing for 30 min, the toluene layer was collected and its absorbance was measured at 520 nm. The Pro content was calculated and expressed as μg g^−1^ FW [21].

### 4.6. Antioxidant Enzyme Activity

A 0.5 g of leaves was homogenized in 1 mL of ice-cold potassium–phosphate buffer (50 mM, pH 7.0) containing potassium chloride (100 mM), ascorbate (1 mM), β-mercaptoethanol (5 mM), and glycerol (10%, *w*/*v*) using a precooled mortar and pestle. Homogenates were centrifuged at 12,000 rpm (Centrifuge 5425/5425 R; Eppendorf, Germany) for 10 min, and supernatants were stored at 4°C for measuring four antioxidant enzyme activities [56]. A 20 μL aliquot of enzyme extract plus 0.3 mL riboflavin was added to 2.5 mL reaction solution [0.3 mL methionine (Met, 100 mM); 1.5 mL phosphate buffer (pH 7.8); 0.3 mL p–nitro–blue tetrazolium chloride (NBT, 750 μM); 0.3 mL ethylene diamine tetraacetic acid (EDTA, 1.0 mM)] in a test tube. Tubes were immediately placed under a 4000 LX fluorescent lamp for the light reduction reaction. After 20 min, the reaction was stopped by shading with black paper. Phosphoric buffer was used as a zero point, and absorption was compared at 560 nm to measure SOD activity. A 0.1 mL aliquot of enzyme extract was mixed thoroughly with 2.9 mL reaction solution [2.6 mL guaiacol (0.3%, *v*/*v*); 0.3 mL H_2_O_2_ (0.6%, *v*/*v*)] in a small tube. Absorption was measured at 470 nm using pH 7.0 phosphoric buffer as a zero point to calculate POD activity. A 0.2 mL aliquot of enzyme extract was added to 2.8 mL H_2_O_2_ (0.067 M) for monitoring the decrease of absorbance at 240 nm for 1 min caused by the decomposition of H_2_O_2_. CAT activity was calculated subsequently. A 0.1 mL aliquot of enzyme extract was mixed thoroughly with 2.9 mL reaction solution [1.7 mL phosphate acid buffer (containing 0.1 mM EDTA, pH 7.0); 0.1 mL ascorbic acid (AsA, 5 mM); 0.1 mL H_2_O_2_ (20 mM)]. Then APX activity was measured by observing the decrease in absorbance at 290 nm.

### 4.7. PAs Concentration

A 0.1 g of leaves was homogenized with 1 mL of HClO_4_ (5%, *w*/*v*), kept in an ice bath for 1 h, and then centrifuged at 12,000 rpm (Centrifuge 5425/5425 R; Eppendorf, Germany) for 30 min at 4 °C. The supernatant was stored at –70 °C for PAs concentration measurements by high-performance liquid chromatography (HPLC; Shimadzu LCMS8040 system, China) [11]. Standard Put (CAS No.: 333–93–7), Spm (CAS No.: 71–44–3), and Spd (CAS No.: 334–50–9) were purchased from Sigma–Aldrich (USA) and were used to optimize the mass spectrometric parameters and fragment spectra [12,13,14].

### 4.8. Chlorophyll Content and Photosynthetic Parameters

A 0.1 g of leaves were digested in 5 mL alcohol (95%, *v*/*v*) overnight at 4 °C, and then centrifuged at 12,000 rpm (Centrifuge 5425/5425 R; Eppendorf, Germany) for 5 min at 4 °C. The supernatant was stored at 4 °C for Chl a, Chl b, Chl a+b, and Chl a/b ratio measurements [46]. Leaf photosynthetic parameters were measured between 10:30 and 11:30 AM using an LI-6400 XT portable photosynthesis system (LI-COR Inc., Lincoln, NE, USA). The Pn, Ci, Gs, and Tr were measured in a chamber at 1,500 μM m^−2^ s^−1^ photosynthetically active radiation (PAR) and 380 ± 5 μM CO_2_ M^−1^ [15]. The WUE and Rubisco activity [47] were estimated as follows:WUE = Pn/Tr(3)
Rubisco = Pn/Ci(4)

### 4.9. RT-qPCR Analysis

Among 39 maize genotypes, we screened out the strongest cold–resistant M0504 and the most cold–sensitive M1012 (Figure 4b). Seedlings of these two genotypes were grown for 7 days at 10 °C and 22 °C, and total RNAs were extracted from leaves with TRIZOL reagent (Invitrogen, Carlsbad, CA, USA). Extracted RNA was reverse–transcribed into cDNA using a SuperScript III First strand Kit (Invitrogen). RT–qPCR was conducted using TransStart Tip Green qPCR SuperMix (Tran, Beijing, China). Primers (Table 2) for six candidate genes involving in PAs metabolisms and antioxidant enzymes system were designed according to previous studies [12,21]. The positions of these genes were mapped in the *Zea_mays* B73_V4 reference genome (https://www.maizegdb.org/, accessed on 11 June 2022), and their functional annotation was performed using the tool AgBase v2.00 (https://agbase.arizona.edu/, accessed on 11 June 2011). Relative gene expression levels were calculated by the 2^−∆∆Ct^ method, with *GRMZM2G126010* as an internal reference gene [56,57].

### 4.10. Statistical Analysis

For all traits under both temperature treatments, we used the IBM-SPSS Statistics v19.0 (SPSS Inc., https://www.Ibm.com/products/spss-statistics, accessed on 11 January 2022) to perform ANOVA among the two variables (maize genotypes [G] and temperatures [T]) and their G × T interaction, and to perform PCA, respectively. The violin plot, correlation coefficient, and interactive ring correlation diagrams of all traits were prepared using the Genescloud tool, a free online platform for data analysis (https://www.genescloud.cn, accessed on 11 January 2022) [57]. Relative to the normal temperature treatment, the rate change (RC) for each trait of single maize genotype under the low temperature treatment was estimated as follows [57]:RC = (T_M–LT_ − T _M–CK_)/T _M–CK_ × 100%,(5)
where T_M–LT_ and T _M–CK_ were the values of corresponding trait in a genotype under the low temperature and normal temperature treatments, respectively. According to Yan et al. [22], the cold–resistance coefficient (CRC_ij_) of the *i*–th trait for the *j*–th maize genotype was calculated as follows:CRC_ij_ = T_LTij_/T_CKij_(6)
where T_LTij_ and T_CKij_ were the values of the *i*–th trait for the *j*–th maize genotype evaluated under low temperature and normal temperature treatment, respectively. Maize cold–resistance in seedling stage was also evaluated by the membership function value [22]. This methodology gives a comprehensive assessment by suing the membership functions based on the theory of fuzzy mathematics. The membership function of a fuzzy set is a generalization of the indicator function in classical sets; it represents the degree of truth as an extension of valuation. For any set T, a membership function on T is any function from T to the real unit interval [0, 1]. According to the cold resistance coefficient, the modified membership function value of cold resistance (MFVC) was estimated as follows:U_ij_ = (CRC_ij_ − CRC_i-min_)/(CRC_i-max_ − CRC_i-min_)(7)
U_ij_ = 1 − (CRC_ij_ − CRC_i-min_)/CRC_i-max_ − CRC_i-min_)(8)
(9)$$U_{j}=\frac{1}{n}\overset{n}{\underset{i=1}{\sum }}U_{\mathrm{ij}}$$
where U_ij_ was the membership function value of the *i*–th trait for the *j*–th maize genotype for cold resistance; CRC_i-min_ and CRC_i-max_ were the minimum and maximum values of the cold resistance coefficient of the i-th trait, respectively, and when the *i*–th trait was positively or negatively correlated with cold resistance of maize, Equation (7) or Equation (8) was estimated, respectively; *U_j_* was the average value of the membership function of measured traits for the *j*–th maize genotypes for cold resistance. The genetic variation coefficient (CV_g_) of single trait among all maize genotypes under each treatment was calculated as follows:(10)$${\mathrm{CV}}_{g}=\mathrm{SD}/\bar{x}\times 100\%$$
where SD was standard deviation, and $\bar{x}$ was the average value of the single trait among all genotypes under each treatment. Hierarchical clustering analysis (HCA) [48] was performed by MultiExperiment Viewer (Mev 4.9.0; https://sourceforge.net/projects/mev-tm4/, accessed on 20 May 2022).

## 5. Conclusions

In summary, maize is a cold sensitive crop that shows little capacity to acclimate to low growth temperature. In this study, we found that maize seedlings exposed to 10 °C low temperature stress led to notable changes (*p* < 0.01 or *p* < 0.001) in multiple agronomic and physiological traits, and the extent of changes in the abovementioned traits were genotypes dependent (*p* < 0.01 or *p* < 0.001); thereby, different maize genotypes improved cold resistance through different mechanisms according to their genetic constitution. The strong cold-resistant maize genotypes accumulated higher levels of MSI, CAT activity, POD activity, Chl a content, Chl b content, Chl a+b content, Pn, Rubisco activity, Tr, WUE, Put concentration, Spd concentration, Spm concentration, seedling length, seedling fresh weight, and plant biomass, which could be good indicators for screening of cold–resistant genotypes of maize. Gene expression analysis using RT–qPCR also showed that the six candidate genes, i.e., *GRMZM2G059991*, *GRMZM2G089982*, *GRMZM2G088212*, *GRMZM2G396553*, *GRMZM2G120578*, and *GRMZM2G396856* that involved in antioxidant enzymes activity and PAs metabolism, may be used for genetic modification to improve maize cold resistance. Moreover, the seven genotypes, including M0504, F0305, M02021, M1017, F0801, F1230, and lg104-1, with the strongest resistance to low temperature, can be used as parents in breeding programs to develop new maize varieties.

## Figures and Tables

**Figure 1 plants-11-01881-f001:**
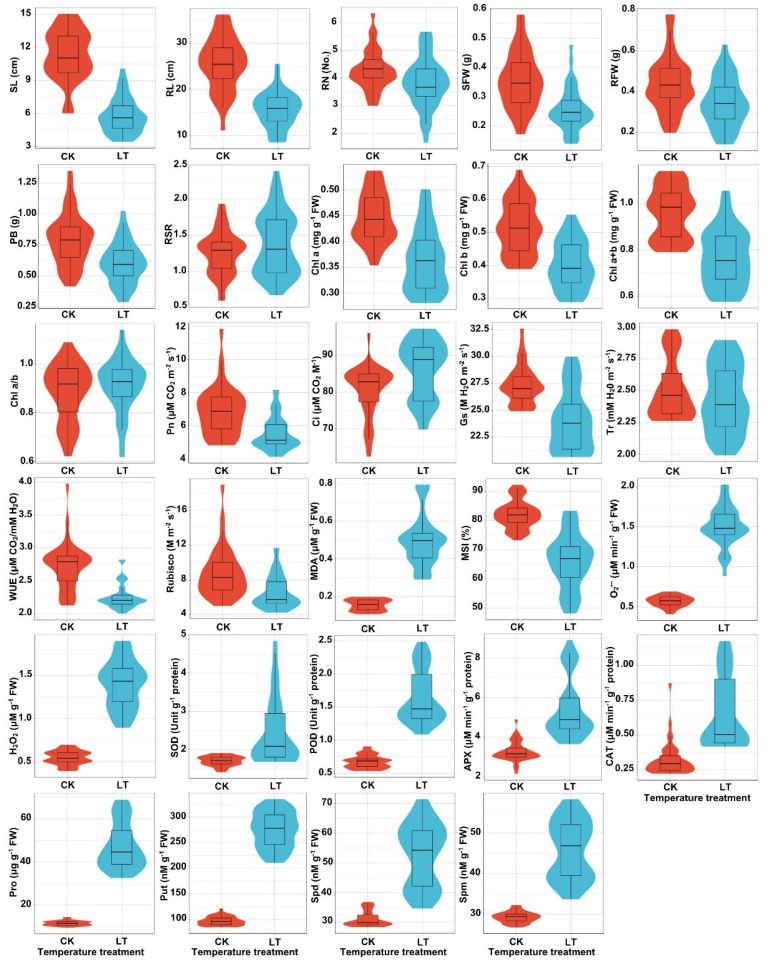
Violin plot of the 29 tested traits in 39 maize genotypes seedlings under 22 °C normal (CK) and 10 °C low temperature (LT) treatments, respectively. The traits included seedling length (SL), root length (RL), root number (RN), seedling fresh weight (SFW), root fresh weight (RFW), plant biomass (PB), root–shoot ratio (RSR), chlorophyll a content (Chl a), chlorophyll b content (Chl b), chlorophyll a+b content (Chl a+b), chlorophyll a:b ratio (Chl a/b), net photosynthetic rate (Pn), intercellular CO_2_ concentration (Ci), stomatal conductance (Gs), transpiration rate (Tr), water use efficiency (WUE), ribulose 1,5–biphosphate carboxylase activity (Rubisco), malondialdehyde content (MDA), membrane stability index (MSI), O_2_^•−^ production rate (O_2_^•−^), H_2_O_2_ content (H_2_O_2_), superoxide dismutase activity (SOD), peroxidase activity (POD), catalase activity (CAT), ascorbate peroxidase activity (APX), proline content (Pro), putrescine concentration (Put), spermidine concentration (Spd), and spermine concentration (Spm), respectively.

**Figure 2 plants-11-01881-f002:**
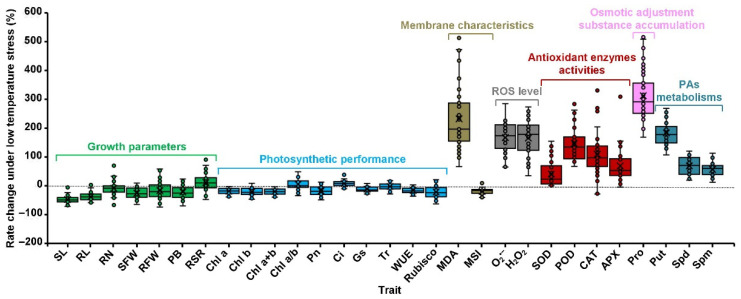
The rate changes among 29 tested traits in 30 maize genotypes seedlings under 22 °C normal (CK) and 10 °C low temperature (LT) treatments. The traits included seedling length (SL), root length (RL), root number (RN), seedling fresh weight (SFW), root fresh weight (RFW), plant biomass (PB), root–shoot ratio (RSR), chlorophyll a content (Chl a), chlorophyll b content (Chl b), chlorophyll a+b content (Chl a+b), chlorophyll a:b ratio (Chl a/b), net photosynthetic rate (Pn), intercellular CO_2_ concentration (Ci), stomatal conductance (Gs), transpiration rate (Tr), water use efficiency (WUE), ribulose 1,5–biphosphate carboxylase activity (Rubisco), malondialdehyde content (MDA), membrane stability index (MSI), O_2_^•−^ production rate (O_2_^•−^), H_2_O_2_ content (H_2_O_2_), superoxide dismutase activity (SOD), peroxidase activity (POD), catalase activity (CAT), ascorbate peroxidase activity (APX), proline content (Pro), putrescine concentration (Put), spermidine concentration (Spd), and spermine concentration (Spm). All these traits were classified into 5 categories including seven growth parameters, ten photosynthetic performances, two membrane characteristics, two reactive oxygen species (ROS), four antioxidant enzymes activities, one osmotic adjustment substance accumulation, and three polyamines (PAs) metabolisms.

**Figure 3 plants-11-01881-f003:**
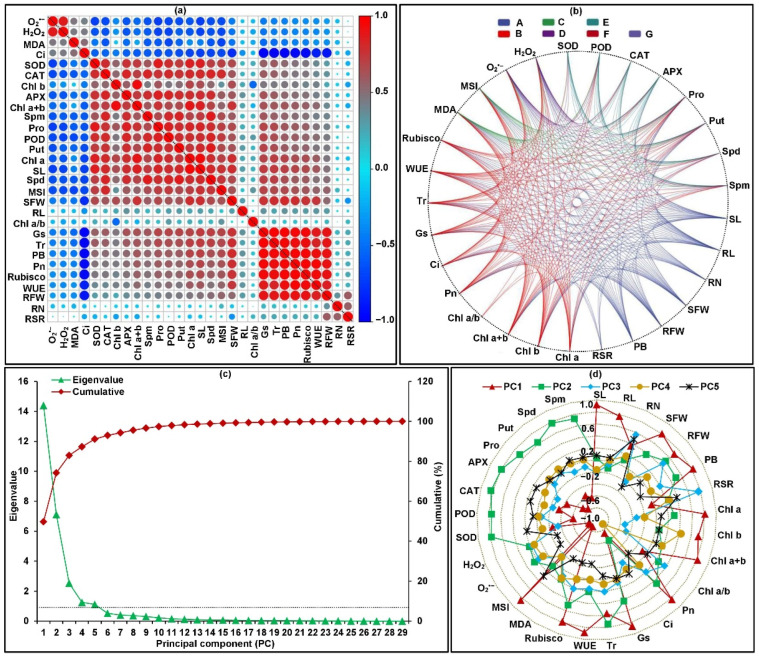
Framework of relations based on correlation and principal component analyses (PCA) of 29 tested traits among 39 maize genotypes seedlings under 22 °C normal and 10 °C low-temperature treatments. The traits included seedling length (SL), root length (RL), root number (RN), seedling fresh weight (SFW), root fresh weight (RFW), plant biomass (PB), root–shoot ratio (RSR), chlorophyll a content (Chl a), chlorophyll b content (Chl b), chlorophyll a+b content (Chl a+b), chlorophyll a:b ratio (Chl a/b), net photosynthetic rate (Pn), intercellular CO_2_ concentration (Ci), stomatal conductance (Gs), transpiration rate (Tr), water use efficiency (WUE), ribulose 1,5–biphosphate carboxylase activity (Rubisco), malondialdehyde content (MDA), membrane stability index (MSI), O_2_^•−^ production rate (O_2_^•−^), H_2_O_2_ content (H_2_O_2_), superoxide dismutase activity (SOD), peroxidase activity (POD), catalase activity (CAT), ascorbate peroxidase activity (APX), proline content (Pro), putrescine concentration (Put), spermidine concentration (Spd), and spermine concentration (Spm), respectively. (**a**) Pearson correlation coefficient diagram among 29 tested traits. (**b**) Interactive ring correlation diagram among all traits. (**c**) Eigenvalues and cumulative variance of principal components (PCs). (**d**) Eigenvectors distributions of 29 traits in PC1, PC2, PC3, PC4, and PC5, respectively.

**Figure 4 plants-11-01881-f004:**
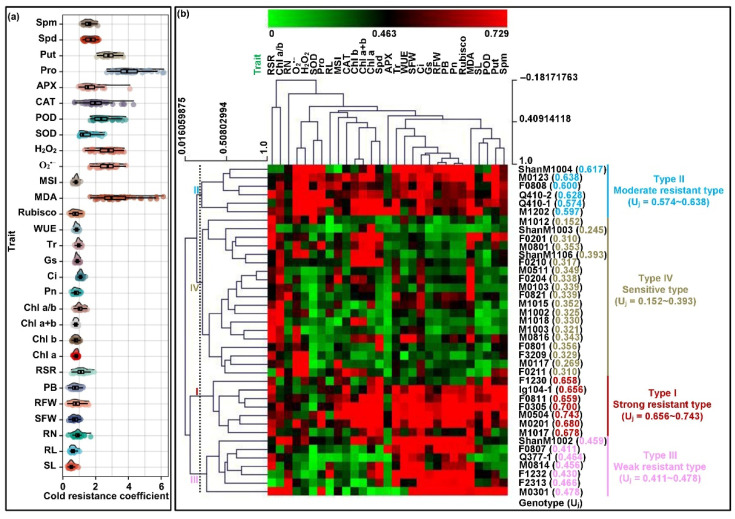
CloudRain of cold resistance coefficient, membership function value and hierarchical clustering analysis (HCA) based on corresponding cold resistance coefficient of 29 tested traits among 39 maize genotypes. The traits included seedling length (SL), root length (RL), root number (RN), seedling fresh weight (SFW), root fresh weight (RFW), plant biomass (PB), root–shoot ratio (RSR), chlorophyll a content (Chl a), chlorophyll b content (Chl b), chlorophyll a+b content (Chl a+b), chlorophyll a:b ratio (Chl a/b), net photosynthetic rate (Pn), intercellular CO_2_ concentration (Ci), stomatal conductance (Gs), transpiration rate (Tr), water use efficiency (WUE), ribulose 1,5–biphosphate carboxylase activity (Rubisco), malondialdehyde content (MDA), membrane stability index (MSI), O_2_^•−^ production rate (O_2_^•−^), H_2_O_2_ content (H_2_O_2_), superoxide dismutase activity (SOD), peroxidase activity (POD), catalase activity (CAT), ascorbate peroxidase activity (APX), proline content (Pro), putrescine concentration (Put), spermidine concentration (Spd), and spermine concentration (Spm), respectively. (**a**) CloudRain of cold resistance coefficient among 29 traits. (**b**) Comprehensive evaluation of cold resistance (U_j_) in 39 maize genotypes using membership function analysis (MFA) and HCA, respectively.

**Figure 5 plants-11-01881-f005:**
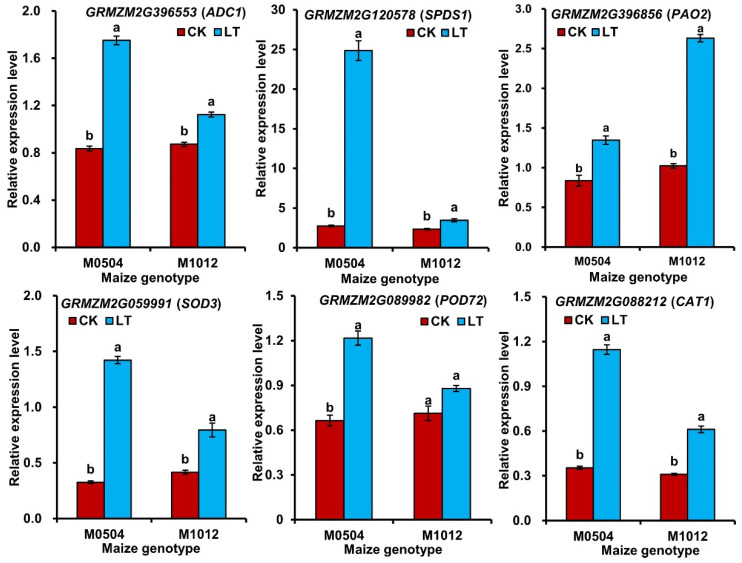
The relative expression of six cold–responsive genes in leaves of two contrasting maize genotypes (M0504, a strong cold–resistant genotype; M1012, a cold–sensitive genotype) under 22 °C normal (CK) and 10 °C low temperature (LT) treatments. The genes included *arginine decarboxylase 1 (ADC1*), *spermidine synthase 1* (*SPDS1*), *probable polyamine oxidase 2* (*PAO2*), *superoxide dismutase 3* (*SOD3*), *peroxidase 72* (*POD72*), and *catalase 1* (*CAT1*), respectively. Data were means of three replicates. Error bars indicated standard error (SD). The different letter indicated significant differences between two treatments within a maize genotypes (*p* < 0.05).

**Figure 6 plants-11-01881-f006:**
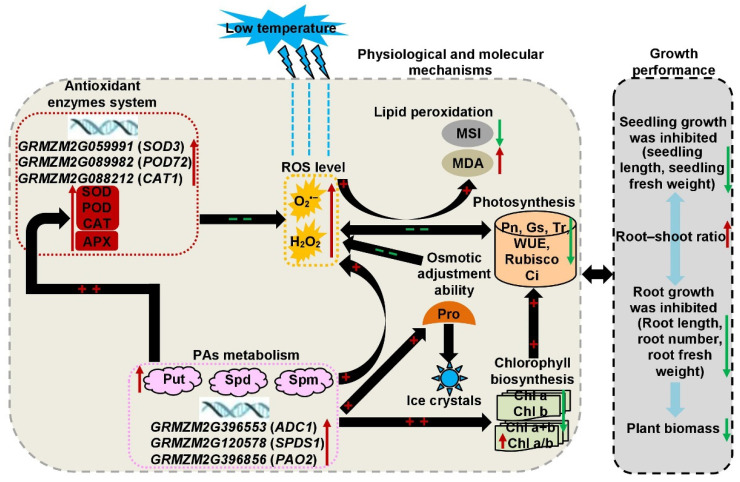
Response mechanisms underlying resistance to low-temperature stress in maize seedlings. The traits included chlorophyll level, i.e., chlorophyll a content (Chl a), chlorophyll b content (Chl b), chlorophyll a+b content (Chl a+b), chlorophyll a:b ratio (Chl a/b); photosynthetic performance, i.e., net photosynthetic rate (Pn), intercellular CO_2_ concentration (Ci), stomatal conductance (Gs), transpiration rate (Tr), water use efficiency (WUE), ribulose 1,5–biphosphate carboxylase activity (Rubisco); antioxidant enzymes system, i.e., superoxide dismutase activity (SOD), peroxidase activity (POD), catalase activity (CAT), ascorbate peroxidase activity (APX); reactive oxygen species (ROS) level, i.e., O_2_^•−^ production rate (O_2_^•−^), H_2_O_2_ content (H_2_O_2_), membrane system, malondialdehyde content (MDA), membrane stability index (MSI); osmotic adjustment ability, i.e., proline content (Pro); and polyamines (PAs) metabolism, i.e., putrescine concentration (Put), spermidine concentration (Spd), and spermine concentration (Spm). The red and green arrow indicated that the corresponding traits were increased and decreased under low-temperature treatment, respectively. “+ +” and “− −” indicated that the corresponding pathways acted positively and negatively, respectively.

**Table 1 plants-11-01881-t001:** Factorial ANOVA of the two examined variables (maize genotypes and temperature), their interactions, and genetic variation coefficient (CV_g_) on 29 tested traits in 39 maize genotypes seedlings under two temperature treatments.

Trait	Genotypes (G)	Temperature (T)	G × T Interaction	CV_g_ (CK) (%)	CV_g_ (LT) (%)
Growth parameter					
SL	*F* = 13.449 (*p* < 0.001)	*F* = 1335.996 (*p* < 0.001)	*F* = 4.637 (*p* < 0.001)	20.58	27.56
RL	*F* = 11.824 (*p* < 0.001)	*F* = 623.327 (*p* < 0.001)	*F* = 3.577 (*p* < 0.001)	22.55	24.66
RN	*F* = 4.237 (*p* < 0.001)	*F* = 17.938 (*p* < 0.001)	*F* = 2.409 (*p* < 0.001)	16.96	23.99
SFW	*F* = 14.358 (*p* < 0.001)	*F* = 241.485 (*p* < 0.001)	*F* = 3.906 (*p* < 0.001)	27.29	28.38
RFW	*F* = 9.652 (*p* < 0.001)	*F* = 67.389 (*p* < 0.001)	*F* = 3.621 (*p* < 0.001)	21.08	34.54
PB	*F* = 1297.621 (*p* < 0.001)	*F* = 211.363 (*p* < 0.001)	*F* = 13.034 (*p* < 0.01)	26.54	27.92
RSR	*F* = 7.794 (*p* < 0.001)	*F* = 15.104 (*p* < 0.001)	*F* = 2.318 (*p* < 0.001)	24.15	31.21
Photosynthetic performance					
Chl a	*F* = 623.203 (*p* < 0.001)	*F* = 182.376 (*p* < 0.001)	*F* = 12.377 (*p* < 0.01)	10.88	16.75
Chl b	*F* = 1025.780 (*p* < 0.001)	*F* = 481.451 (*p* < 0.001)	*F* = 4.053 (*p* < 0.05)	15.46	17.90
Chl a+b	*F* = 932.963 (*p* < 0.001)	*F* = 365.686 (*p* < 0.001)	*F* = 2.760 (*p* < 0.05)	11.49	16.07
Chl a/b	*F* = 146.129 (*p* < 0.001)	*F* = 172.080 (*p* < 0.001)	*F* = 30.104 (*p* < 0.001)	13.45	12.02
Pn	*F* = 28.235 (*p* < 0.001)	*F* = 9.676 (*p* < 0.001)	*F* = 2.172 (*p* < 0.001)	21.44	18.63
Ci	*F* = 163.418 (*p* < 0.001)	*F* = 13.466 (*p* < 0.001)	*F* = 9.302 (*p* < 0.001)	8.67	9.21
Gs	*F* = 210.699 (*p* < 0.001)	*F* = 78.650 (*p* < 0.001)	*F* = 4.512 (*p* < 0.05)	6.52	11.82
Tr	*F* = 99.425 (*p* < 0.001)	*F* = 38.087 (*p* < 0.001)	*F* = 9.110 (*p* < 0.001)	9.19	10.87
WUE	*F* = 98.802 (*p* < 0.001)	*F* = 38.862 (*p* < 0.001)	*F* = 9.176 (*p* < 0.001)	13.16	8.43
Rubisco	*F* = 78.860 (*p* < 0.001)	*F* = 22.607 (*p* < 0.001)	*F* = 7.635 (*p* < 0.001)	32.15	29.16
Membrane characteristics					
MDA	*F* = 311.124 (*p* < 0.001)	*F* = 1518.619 (*p* < 0.001)	*F* = 153.605 (*p* < 0.001)	18.72	27.18
MSI	*F* = 340.754 (*p* < 0.001)	*F* = 387.618 (*p* < 0.001)	*F* = 30.448 (*p* < 0.001)	5.76	14.07
ROS level					
O_2_^•−^	*F* = 1350.626 (*p* < 0.001)	*F* = 6675.706 (*p* < 0.001)	*F* = 513.397 (*p* < 0.001)	11.82	16.49
H_2_O_2_	*F* = 1125.0331 (*p* < 0.001)	*F* = 5655.161 (*p* < 0.001)	*F* = 320.381 (*p* < 0.001)	15.20	19.74
Antioxidant enzymes activity					
SOD	*F* = 2457.673 (*p* < 0.001)	*F* = 3069.115 (*p* < 0.001)	*F* = 1400.441 (*p* < 0.001)	7.43	34.85
POD	*F* = 9.451 (*p* < 0.01)	*F* = 56.748 (*p* < 0.001)	*F* = 2.291 (*p* < 0.05)	14.56	24.27
CAT	*F* = 1023.800 (*p* < 0.001)	*F* = 360.449 (*p* < 0.001)	*F* = 10.329 (*p* < 0.01)	37.59	40.13
APX	*F* = 6469.477 (*p* < 0.001)	*F* = 7908.238 (*p* < 0.001)	*F* = 700.159 (*p* < 0.001)	15.35	27.81
Osmotic adjustment substance accumulation					
Pro	*F* = 138.342 (*p* < 0.001)	*F* = 10.049 (*p* < 0.01)	*F* = 5.876 (*p* < 0.01)	9.69	22.48
PAs metabolism					
Put	*F* = 555.242 (*p* < 0.001)	*F* = 32.162 (*p* < 0.01)	*F* = 11.102 (*p* < 0.001)	8.87	12.72
Spd	*F* = 78.228 (*p* < 0.001)	*F* = 11.436 (*p* < 0.01)	*F* = 7.118 (*p* < 0.001)	8.32	20.02
Spm	*F* = 35.699 (*p* < 0.001)	*F* = 31.052 (*p* < 0.001)	*F* = 15.870 (*p* < 0.001)	4.59	15.62

SL, seedling length; RL, root length; RN, root number; SFW, seedling fresh weight; RFW, root fresh weight; PB, plant biomass; RSR, root–shoot ratio; Chl a, chlorophyll a content; Chl b, chlorophyll b content; Chl a+b, chlorophyll a+b content; Chl a/b, chlorophyll a:b ratio; Pn, net photosynthetic rate; Ci, intercellular CO_2_ concentration; Gs, stomatal conductance; Tr, transpiration rate; WUE, water use efficiency; Rubisco, ribulose 1,5–biphosphate carboxylase activity; MDA, malondialdehyde content; MSI, membrane stability index; O_2_^•−^, O_2_^•−^ production rate; H_2_O_2_, H_2_O_2_ content; SOD, superoxide dismutase activity; POD, peroxidase activity; CAT, catalase activity; APX, ascorbate peroxidase activity; Pro, proline content; Put, putrescine concentration; Spd, spermidine concentration; Spm, spermine concentration, ROS, reactive oxygen species; PAs, polyamines; CV_g_ (CK) and CV_g_ (LT) were the genetic variation coefficient of corresponding traits under 22 °C normal (CK) and 10 °C low temperature (LT) treatments, respectively.

**Table 2 plants-11-01881-t002:** Sequences of primers used in reverse–transcription quantitative PCR (RT–qPCR) and functional annotation of six candidate genes.

Gene ID (Encoded Protein)	Gene Position	Primer Sequence (5′ to 3′)	Gene Functional Annotation
*GRMZM2G396553* (*Arginine decarboxylase 1*, *ADC1*)	Chromosome 9 (23396672_23399720 bp)	F: GCTACGGCTCAAGGTACCAG R: CCGAACTCCACAATGTCCTC	Arginine decarboxylase activity (GO:0008792); Arginine catabolic process (GO:0006527); Response to cold (GO:0009409)
*GRMZM2G120578* (*Spermidine synthase 1*, *SPDS1*)	Chromosome 5 (72695353_72701535 bp)	F: TGTTCAATTCCATCCCCTAAA R: TCCACTGAACTGTGTCTGGAA	Catalytic activity (GO:0003824); Transferase activity (GO:0016740); Polyamine metabolic process (GO:0006595); Polyamine biosynthetic process (GO:0006596)
*GRMZM2G396856* (*Probable polyamine oxidase 2*, *PAO2*)	Chromosome 10 (143652563_143657803 bp)	F: CACACACACCATCCGCTATT R: CATCAGCAGCAGCAAGACC	Oxidoreductase activity (GO:0016491); Polyamine oxidase activity (GO:0046592); Polyamine catabolic process (GO:0006598)
*GRMZM2G059991* (*Superoxide dismutase 3*, *SOD3*)	Chromosome 6 (140001982_140006117 bp)	F: TCACCCAAGAGGGAGATG R: TTGCTCGCAGGATTGTAG	Removal of superoxide radicals (GO:0019430); Superoxide dismutase activity (GO:0004784); Oxidoreductase activity (GO:0016491)
*GRMZM2G089982* (*Peroxidase 72*, *POD72*)	Chromosome 3 (40090724_40092823 bp)	F: GGATGTATCCTACGCCGCAA R: TTGTCAAACTTGGCAGGGGT	Peroxidase activity (GO:0004601); Oxidoreductase activity (GO:0016491); Response to oxidative stress (GO:0006979)
*GRMZM2G088212* (*Catalase 1*, *CAT1*)	Chromosome 5 (65456774_65461269 bp)	F: CCGAATCCAAAGACCAAT R: ATGCCAACATCGTCAAAGAG	Catalase activity (GO:0004096); Peroxidase activity (GO:0004601); Response to oxidative stress (GO:0006979); Response to heat (GO:0009408)
*GRMZM2G126010* (*actin 1*)	Chromosome 8 (102413768_102417536 bp)	F: CGATTGAGCATGGCATTGTCA R: CCCACTAGCGTACAACGAA	ATP binding (GO:0005524); Nucleotide binding (GO:0000166)

## Data Availability

Data is contained within the article.

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
