# Peer review of "Understanding and Comprehensive Evaluation of Cold Resistance in the Seedlings of Multiple Maize Genotypes"

_plants, 2022, doi:10.3390/plants11141881_

Round 1

Reviewer 1 Report

The authors of this article screened 36 maize lines for their reaction to cold temperature by treating them under 10 C low temperature and 22 C normal temperature conditions for 7 days and assessed several agronomic and physiological traits. Thy further discussed the changes in these agronomic and physiological traits and how they are genotype dependent. They further studied the differential expression of six candidate genes involved in ROS scavenging and polyamines (PA) metabolism on two extreme genotypes based on their cold sensitivity or resistance.

This study provides some basic and initial information on how some of the maize lines react to cold stress. This study has a potential to be extended to a larger maize panel or multiple breeding lines to select better lines for the purpose of understanding cold sensitivity as well as their use in breeding program. However, I have a couple of questions and suggestions:

Major:

·         Materials and methods lack essential information like, how exactly MSI, MDA content, ROS levels, Pro content and antioxidant enzyme activity was assessed. I see citations but at least in brief it should be mentioned how exactly they were measured, what methods or equipment used for the purpose.

·         The authors in the section 4.9 RT-qPCR Analysis, explained that positions of the selected candidate genes were mapped to the B73_V4 but the gene model presented are of either V2 or 3. I suggest rechecking the gene positions and including gene model ID from V4 as well.

Minor:

·         Line 52-54: “….resulting in decrease the electron transport…” reads odd, I think you mean “…resulting in decreased or reduced the electron transport..”?

·         Line 82-83: “…beneficial for their elite allele application.” Application in what?

·         Table 1: I suggest adding one more column with the category each phenotype falls in. This will help readers.

·         I think some representative pictures of plants under stress conditions should be included in the manuscript. At least pictures of the extreme genotypes should be added.

·         There are a few more grammatical errors that must be looked into.

Author Response

Dear Ms. Sasithorn Sukkleang and Reviewers

Thank you for your letter of – and for the referee’s comments concerning our manuscript, “Understanding and Comprehensive Evaluation of Cold Resistance in the Seedlings of Multiple Maize Genotypes (Manuscript ID: plants-1813004)”. We have carefully studied these comments and have made corresponding corrections to the manuscript, which we describe in detail below. We would like to re-submit the manuscript and that for possible publication on the Special Issue “Plant and Microbe Adaptations to Cold” of Plants. Thank you very much for your time and consideration.

Ms. Sasithorn Sukkleang of Assistant Editor

Your manuscript has been reviewed by experts in the field. Please find your manuscript with the referee reports at this link: https://susy.mdpi.com/user/manuscripts/resubmit/929c2e7234dbb18dd4f85d67996e678f

Thanks for the positive comments of you and all reviewers for our manuscript. As suggested, we have downloaded the referee reports and carefully revised the manuscript, and all corresponding contents were modified by “Track Changes” function of MS Word. We then have re-submitted the manuscript.

Thank you for your time and consideration.

 (I) Please revise your manuscript according to the referees’ comments and upload the revised file within 5 days.

Thanks for the positive comments of you and all reviewers for our manuscript. As suggested, we have carefully revised the manuscript. We then have re-submitted the manuscript.

Thank you for your time and consideration.

(II) Please use the version of your manuscript found at the above link for your revisions.

Thanks for the positive comments of you and all reviewers for our manuscript. As suggested, we have found our manuscript at the above link and carefully revised the manuscript in turn. We then have re-submitted the manuscript.

Thank you for your time and consideration.

(III) Please check that all references are relevant to the contents of the manuscript.

Thanks for the positive comments of you and all reviewers for our manuscript. As suggested, we have carefully checked all references to make sure they are necessary. Moreover, in order to better discuss with our results in this study, we have re-added three references in Lines 749-750 and 776-778 of the manuscript. In addition, we have also revised the errors of references in References section of the manuscript (Lines 681-808). We then have re-submitted the manuscript.

Thank you for your time and consideration.

(IV) Any revisions made to the manuscript should be marked up using the “Track Changes” function if you are using MS Word/LaTeX, such that changes can be easily viewed by the editors and reviewers.

Thanks for the positive comments of you and all reviewers for our manuscript. As suggested, we have carefully revised the manuscript, and all corresponding contents were modified by “Track Changes” function of MS Word. We then have re-submitted the manuscript.

Thank you for your time and consideration.

(V) Please provide a short cover letter detailing your changes for the editors’ and referees’ approval.

Thanks for the positive comments of you and all reviewers for our manuscript. As suggested, we have carefully revised the manuscript. In addition, we have prepared a detailed response letter to you and all reviewers for each point.  We then have re-submitted the manuscript.

Thank you for your time and consideration.

If one of the referees has suggested that your manuscript should undergo extensive English revisions, please address this issue during revision. We propose that you use one of the editing services listed at https://www.mdpi.com/authors/english or have your manuscript checked by a native English-speaking colleague.

Thanks for the positive comments you and all reviewers for our manuscript. The English language of the manuscript has been well modified by the professor Wun Chao (United States Department of Agriculture, Fargo, ND 58102-2765, USA; wun.chao@usda.gov) and the Charlesworth Author Services (https://www.cwauthors.com.cn/). We then have re-submitted the manuscript.

Thank you for your time and consideration.

Please do not hesitate to contact us if you have any questions regarding the revision of your manuscript or if you need more time. We look forward to hearing from you soon.

Thanks for the positive comments you and all reviewers for our manuscript.

Thank you for your time and consideration.

Reviewer 1

The authors of this article screened 36 maize lines for their reaction to cold temperature by treating them under 10 C low temperature and 22 C normal temperature conditions for 7 days and assessed several agronomic and physiological traits. Thy further discussed the changes in these agronomic and physiological traits and how they are genotype dependent. They further studied the differential expression of six candidate genes involved in ROS scavenging and polyamines (PA) metabolism on two extreme genotypes based on their cold sensitivity or resistance. This study provides some basic and initial information on how some of the maize lines react to cold stress. This study has a potential to be extended to a larger maize panel or multiple breeding lines to select better lines for the purpose of understanding cold sensitivity as well as their use in breeding program. However, I have a couple of questions and suggestions:

Thanks for your positive comments. In addition, as suggested, we have further revised and answered all questions for the manuscript. We then have re-submitted the manuscript.

Thank you for your time and consideration.

Major:

Materials and methods lack essential information like, how exactly MSI, MDA content, ROS levels, Pro content and antioxidant enzyme activity was assessed. I see citations but at least in brief it should be mentioned how exactly they were measured, what methods or equipment used for the purpose.

Thanks for your positive comments. As suggested, we have re-added the detailed determination methods for these traits in Materials and Methods section of the manuscript. Such as:

MSI: “A 0.1 g of leaves was placed in test tubes containing 10 mL of double–distilled water and incubated at 40°C for 30 min. The conductivity (C1) was measured using a DDSJ-308F conductivity meter (Rex Electric Chemical, China). The same set was then kept in a water bath at 100°C for 15 min, and the conductivity was recorded (C2). The MSI [20] was calculated as follows: MSI = [1-(C1/C2)] × 100 % (2)” in Lines 525-529 of the manuscript. We then have re-submitted the manuscript.

MDA content: “The lipid peroxidation was measured by MDA content [52]. Briefly, a 0.5 g of leaves was homogenized in 5 mL of 0.3% thiobarbituric acid (TBA) and 10% (v/v) trichloroacetic acid. After incubation at 100°C for 30 min, mixtures were centrifuged at 12,000 rpm (Centrifuge 5425/5425 R; Eppendorf, Germany) for 10 min. The absorbance of the colored supernatant was measured at 450 nm, 532 nm, and 600 nm, respectively. Then MDA concentration was calculated and its content was expressed as μM g−1 fresh weight (FW).” in Lines 530-535 of the manuscript. We then have re-submitted the manuscript.

ROS level: “A 0.5 g of leaves was homogenized in 6 mL of phosphate buffer (65 mM, pH 7.8) and then centrifuged at 10,000 rpm (Centrifuge 5425/5425 R; Eppendorf, Germany) at 4°C for 10 min. Next, 1 mL of supernatant, 0.9 mL of phosphate buffer (65 mM, pH 7.8), and 0.1 mL of hydroxylamine hydrochloride (10 mM) were mixed, and the solutions were placed in a water bath for 20 min at 25°C. To 0.5 mL of those solutions, 0.5 mL of sulfanilic acid (17 mM) and 0.5 mL of α-naphthylamine (17 mM) were added, followed by 20 min of incubation at 25°C in a water bath. O2•− production rate was determined following the addition of N-butyl alcohol and read at 530 nm, which was expressed as μM min−1 g−1 FW [53]. A 0.5 g of leaves was homogenized in 5 mL of ice–cold trichloroacetic acid (0.1%, w/v) and centrifuged at 12,000 rpm (Centrifuge 5425/5425 R; Eppendorf, Germany) at 4°C for 15 min to collect the supernatant, then optical absorption of the supernatant was measured spectrophotometrically at 410 nm to measure H2O2 content and expressed as μM g−1 FW [54].” in Lines 537-549 of the manuscript. We then have re-submitted the manuscript.

Pro content: “A 0.5 g of leaves was homogenized in 10 mL of 3% ice–cold sulfosalicylic acid and then centrifuged at 12,000 rpm (Centrifuge 5425/5425 R; Eppendorf, Germany) at 4°C for 15 min. Supernatant, 2 mL, was mixed with 2 mL of acid ninhydrin and 2 mL of glacial acetic acid. After incubation at 100°C for 1 h, 4 mL of toluene was added to extract red substance. After standing for 30 min, the toluene layer was collected and its absorbance was measured at 520 nm. The Pro content was calculated and expressed as μg g−1 FW [21].” in Lines 551-557 of the manuscript. We then have re-submitted the manuscript.

Antioxidant enzyme activity: “A 0.5 g of leaves was homogenized in 1 mL of ice–cold potassium–phosphate buffer (50 mM, pH 7.0) containing potassium chloride (100 mM), ascorbate (1 mM), β-mercaptoethanol (5 mM), and glycerol (10%, w/v) using a precooled mortar and pestle. Homogenates were centrifuged at 12,000 rpm (Centrifuge 5425/5425 R; Eppendorf, Germany) for 10 min, and supernatants were stored at 4°C for measuring four antioxidant enzyme activities [55]. A 20 μL aliquot of enzyme extract plus 0.3 mL riboflavin was added to 2.5 mL reaction solution [0.3 mL methionine (Met, 100 mM); 1.5 mL phosphate buffer (pH 7.8); 0.3 mL p–nitro–blue tetrazolium chloride (NBT, 750 μM); 0.3 mL ethylene diamine tetraacetic acid (EDTA, 1.0 mM)] in a test tube. Tubes were immediately placed under a 4000 LX fluorescent lamp for light reduction reaction. After 20 min, the reaction was stopped by shading with black paper. Phosphoric buffer was used as a zero point, and absorption was compared at 560 nm to measure SOD activity. A 0.1 mL aliquot of enzyme extract was mixed thoroughly with 2.9 mL reaction solution [2.6 mL guaiacol (0.3%, v/v); 0.3 mL H2O2 (0.6%, v/v)] in a small tube. Absorption was measured at 470 nm using pH 7.0 phosphoric buffer as a zero point to calculate POD activity. A 0.2 mL aliquot of enzyme extract was added to 2.8 mL H2O2 (0.067 M) for monitoring the decrease of absorbance at 240 nm for 1 min caused by the decomposition of H2O2. CAT activity was calculated subsequently. A 0.1 mL aliquot of enzyme extract was mixed thoroughly with 2.9 mL reaction solution [1.7 mL phosphate acid buffer (containing 0.1 mM EDTA, pH 7.0); 0.1 mL ascorbic acid (AsA, 5 mM); 0.1 mL H2O2 (20 mM)]. Then APX activity was measured by observing the decrease in absorbance at 290 nm.” in Lines 559-579 of the manuscript. We then have re-submitted the manuscript.

Thank you for your time and consideration.

The authors in the section 4.9 RT-qPCR Analysis, explained that positions of the selected candidate genes were mapped to the B73_V4 but the gene model presented are of either V2 or 3. I suggest rechecking the gene positions and including gene model ID from V4 as well.

Thanks for your positive comments. As suggested, we have carefully checked and revised the information of the selected candidate genes to make sure that were mapped to the B73_V4 reference genome (https://www.maizegdb.org/). Thereby, we have revised the Table 2 (Lines 610-611) and then have re-submitted the manuscript.

Thank you for your time and consideration.

Minor:

Line 52-54: “….resulting in decrease the electron transport…” reads odd, I think you mean “…resulting in decreased or reduced the electron transport..”?

Thanks for your positive comments. Yes, I totally agree with you, and the corresponding content has been revised that “ROS production also damaged the chloroplast, mitochondria and metabolic enzymes [4,5]. The pale green appearance caused by low levels of chlorophyll (Chl) a and Chl b, resulting in decreased the electron transport in the photosystems, the activity of the enzymes involved in carbon fixation including 1,5–biphosphate carboxylase (Rubisco), NADP-malic enzyme (NADP–ME), pyruvate phosphate dikinase (PPDK), and phosphoenolpyruvate carboxykinase (PEPCK), and net photosynthesis efficiency [5].” in Lines 56-62 of the manuscript. We then have re-submitted the manuscript.

Thank you for your time and consideration.

Line 82-83: “…beneficial for their elite allele application.” Application in what?

Thanks for your positive comments. As suggested, the corresponding content has been revised that “In addition, evaluating the physiological and biochemical changes occurring under low temperature stress may lead to the genetic improvement of cold–resistant genotypes and be beneficial for their elite allele exploration and application in germplasm resources [16,17].” in Lines 92-95 of the manuscript. We then have re-submitted the manuscript.

Thank you for your time and consideration.

 Table 1: I suggest adding one more column with the category each phenotype falls in. This will help readers.

Thanks for your positive comments. As suggested, we have re-added the category of each trait in Table 1 (Lines 117-130). We have re-submitted the manuscript.

Thank you for your time and consideration.

I think some representative pictures of plants under stress conditions should be included in the manuscript. At least pictures of the extreme genotypes should be added.

Thanks for your positive comments. Yes, I totally agree with you. In this study, due to the large number of experimental materials (including 39 maize genotypes), experimental treatments (including 22°C normal and 10°C low temperature treatments), and tested traits (totally 29 tested traits including seven growth parameters, two membrane characteristics, two ROS levels, four antioxidant enzymes activities, ten photosynthetic performances, one osmotic adjustment substance accumulation, and three PAs metabolisms). Moreover, multiple traits need to be determined in a short time with fresh samples under different temperature treatments, and in order to obtain the accurate measurement results of these traits, our focus was to quickly determine the corresponding traits of 39 maize genotypes under two temperature treatments, so we neglected to collect representative or extreme pictures of maize genotypes under both treatments in time. In addition, we have accurately observed seven seedling phenotypes of 39 maize genotypes under both temperature treatments in this study, namely, seedling length, root length, root number, seedling fresh weight, root fresh weight, plant biomass, and root-shoot ratio, these seedling phenotypes can objectively and comprehensively reflect the growth of corresponding maize genotypes under different environments.

Thank you for your time and consideration.

There are a few more grammatical errors that must be looked into.

Thanks for your positive comments. As suggested, we have carefully revised the grammatical errors. In addition, the English language of the manuscript has been well modified by the professor Wun Chao (United States Department of Agriculture, Fargo, ND 58102-2765, USA; wun.chao@usda.gov) and the Charlesworth Author Services (https://www.cwauthors.com.cn/). We then have re-submitted the manuscript.

Thank you for your time and consideration.

Reviewer 2

Comments and Suggestions for Authors:

Authors, identified seven strong cold-resistant genotypes and they can be used as parents in maize breeding programs to develop new varieties.

Thanks for your positive comments.

Thank you for your time and consideration.

My comments are below:

The abstract must include data regarding the critical finds by the authors in terms of data of important findings. 

Thanks for your positive comments. As suggested, we have re-written the abstract to make sure that the abstract including data regarding the critical finds, i.e., “Maize is a cold sensitive crop, and it exhibits severe retardation of growth and development when exposed to cold snaps during and right after seedling emergence. Although different agronomic, physiological, and molecular approaches have been tried to overcome the problems related to cold stress in recent years, the mechanisms causing cold resistance in maize is still unclear. Screening and breeding of varieties for cold resistance may be a sustainable option to boost maize production under low temperature environments. Herein, seedlings of 39 different maize genotypes were treated under both 10°C low temperature and 22°C normal temperature conditions for 7 days, to assess the changes in seven growth parameters, two membrane characteristics, two reactive oxygen species (ROS) levels and four antioxidant enzymes activities. The changes of ten photosynthetic performances, one osmotic substance accumulation, and three polyamines (PAs) metabolisms were also measured. Results indicated that significant differences among genotypes, temperature treatments, and their interactions were found in 29 studied traits, and cold–stressed seedlings were capable to enhance their cold resistance by maintaining high levels of membrane stability index (66.07%); antioxidant enzymes activities including activity of superoxide dismutase (2.44 Unit g−1 protein), peroxidase (1.65 Unit g−1 protein), catalase (0.65 μM min−1 g−1 protein), and ascorbate peroxidase (5.45 μM min−1 g−1 protein); chlorophyll (Chl) content, i.e., Chl a (0.36 mg g−1 FW) and Chl b (0.40 mg g−1 FW); photosynthetic capacity such as net photosynthetic rate (5.52 μM m−2 s−1) and ribulose 1,5–biphosphate carboxylase activity (6.57 M m−2 s−1); PAs concentration, mainly putrescine (274.89 nM g−1 FW), spermidine (52.69 nM g−1 FW), and spermine (45.81 nM g−1 FW), particularly under extended cold stress. Importantly, 16 traits can be good indicators for screening of cold–resistant genotypes of maize. Gene expression analysis showed that GRMZM2G059991, GRMZM2G089982, GRMZM2G088212, GRMZM2G396553, GRMZM2G120578, and GRMZM2G396856 involved in antioxidant enzymes activity and PAs metabolism, and these genes may be used for genetic modification to improve maize cold resistance. Moreover, seven strong cold–resistant genotypes were identified, and they can be used as parents in maize breeding programs to develop new varieties.” in Lines 12-35 of the manuscript. We then have re-submitted the manuscript.

Thank you for your time and consideration.

The introduction must have a clear hypothesis and significantly develop the second paragraph of this manuscript.

Thanks for your positive comments. As suggested, and based on the previous studies in Lines 52-95 of the manuscript, then we have put forward a clear hypothesis in the Introduction section of the manuscript, i.e. “We hypothesize that (i) low temperature caused maize physiological changes and metabolic disturbances, and (ii) they act synergistically to form maize cold resistance.” in Lines 95-97 of the manuscript. “In the context of these facts, this study aimed to explore the response mechanism on multiple agronomic and physiological traits of 39 elite maize genotypes seedlings under 22°C and 10°C temperature conditions, establish a reliable maize screening method for large–scale selection or for breeding stock with cold–resistance, and further screen maize genotypes for resistance against low temperature stress in spring maize region of Northwest China. In addition, the expression level of associated genes was determined via reverse–transcription quantitative PCR (RT–qPCR). These findings will lay a foundation for develop excellent maize cultivars with cold resistance.” in Lines 97-105. We then have re-submitted the manuscript.

In addition, as suggested, we have significantly develop the second paragraph (Lines 52-81) of our manuscript. We then have re-submitted the manuscript.

Thank you for your time and consideration.

Overall there is the repetition of the information which could be avoided.

Thanks for your positive comments. As suggested, we modified and improved the repetition of the information in the manuscript as much as possible, and the professor Wun Chao (United States Department of Agriculture, Fargo, ND 58102-2765, USA; wun.chao@usda.gov) and the Charlesworth Author Services (https://www.cwauthors.com.cn/) have improved the English language to avoid repetition and check errors. All corresponding contents were modified by “Track Changes” function of MS Word. We then have re-submitted the manuscript.

Thank you for your time and consideration.

Check figure ligands; they are carelessly written.

Thanks for your positive comments. As suggested, we have re-made all Figures and Figure ligands. In addition, we have also improved the title and legend of the Figures. Namely, Figure 1 in Lines 173-185, Figure 2 in Lines 215-230, Figure 3 in Lines 255-270, Figure 4 in Lines 291-306, Figure 5 in Lines 319-327, and Figure 6 in Lines 491-505 of the manuscript. We then have re-submitted the manuscript.

Thank you for your time and consideration.

Discussion should include more information and references related to the relevant and related works. 

Thanks for your positive comments. As suggested, we have re-added three references (Lines 749-750 and 776-778), we further combine our results and previous studies or corresponding references (Lines 681-808) to deeply elaborate the Discussion section of the manuscript (Lines 328-505). We then have re-submitted the manuscript.

Thank you for your time and consideration.

Restructure and carefully edit the conclusion section.

Thanks for your positive comments. As suggested, we have re-edited the Conclusion section of the manuscript, Namely, “In summary, maize is a cold sensitive crop that shows little capacity to acclimate to low growth temperature. In this study, we found that maize seedlings exposed to 10°C low temperature stress led to notable changes (P < 0.01 or P < 0.001) in multiple agronomic and physiological traits, and the extent of changes in abovementioned traits were genotypes dependent (P < 0.01 or P < 0.001); thereby, different maize genotypes improved cold resistance through different mechanisms according to their genetic constitution. The strong cold resistant maize genotypes accumulated the higher levels of MSI, CAT activity, POD activity, Chl a content, Chl b content, Chl a+b content, Pn, Rubisco activity, Tr, WUE, Put concentration, Spd concentration, Spm concentration, seedling length, seedling fresh weight, and plant biomass, which could be good indicators for screening of cold–resistant genotypes of maize. Gene expression analysis using RT–qPCR also showed that the six candidate genes, i.e., GRMZM2G059991, GRMZM2G089982, GRMZM2G088212, GRMZM2G396553, GRMZM2G120578, and GRMZM2G396856 that involved in antioxidant enzymes activity and PAs metabolism, may be used for genetic modification to improve maize cold resistance. Moreover, the seven genotypes, including M0504, F0305, M02021, M1017, F0801, F1230, and lg104-1, with the strongest resistance to low temperature, can be used as parents in breeding programs to develop new maize varieties.” in Lines 648-665. We then have re-submitted the manuscript.

Thank you for your time and consideration.

In addition, the professor Wun Chao (United States Department of Agriculture, Fargo, ND 58102-2765, USA; wun.chao@usda.gov) and Charlesworth Author Services (https://www.cwauthors.com.cn/) for providing linguistic assistance during the preparation of this manuscript, and other any modifications were performed by “Track Changes” function of MS Word.

Thank you for your time and consideration.

Sincerely,

Yining Niu professor

State Key Laboratory of Aridland Crop Science, Gansu Agricultural University

E-mail: niuyn@gsau.edu.cn

Tel: +86-139-1913-0638

Reviewer 2 Report

Authors, identified seven strong cold-resistant genotypes and they can be used as parents in maize breeding programs to develop new varieties.

My comments are below:

  • The abstract must include data regarding the critical finds by the authors in terms of data of important findings. 
  • The introduction must have a clear hypothesis and significantly develop the second paragraph of this manuscript.
  • Overall there is the repetition of the information which could be avoided.  
  •  Check figure ligands; they are carelessly written.
  • Discussion should include more information and references related to the relevant and related works. 
  • Restructure and carefully edit the conclusion section.

Author Response

(The authors gave the same response as above.)
